# Prevalence of diabetic foot at risk of ulcer development and its components stratification according to the international working group on the diabetic foot (IWGDF): A systematic review with metanalysis

**Tania Maldonado-Valer[1], Luis F. Pareja-Mujica[1], Rodrigo Corcuera-Ciudad[2], Fernando Andres Terry-Escalante[3,4], Mylenka Jennifer Chevarría-Arriaga[3], Tery Vasquez-Hassinger[1], Marlon Yovera-Aldana[5]***

**1** Universidad Científica del Sur, Lima, Peru, **2** Universidad Científica del Sur, Facultad de Ciencias de la Salud, Carrera de Medicina Humana, CHANGE Research Working Group, Lima, Peru, **3** Universidad de San Martín de Porres, Facultad de Medicina Humana. Lima, Peru, **4** Red de Eficacia Clínica y Sanitaria (REDECS), Lima, Peru, **5** Grupo de Investigación en Neurociencias, Efectividad Clínica y Salud Pública, Universidad Científica del Sur, Lima, Peru

* myovera@cientifica.edu.pe

## Abstract

### Aims

To determine the overall prevalence of diabetic foot at risk according to the International Working Group on the Diabetic Foot stratification.

### Materials and methods

We searched PubMed/Medline, Scopus, Web of Science, and Embase. We included cross-sectional studies or cohorts from 1999 to March 2022. We performed a meta-analysis of proportions using a random-effects model. We assessed heterogeneity through subgroup analysis by continent and other characteristics.

### Results

We included 36 studies with a total population of 11,850 people from 23 countries. The estimated overall prevalence of diabetic foot at risk was 53.2% (95% CI: 45.1–61.3), I2 = 98.7%, p < 0.001. In the analysis by subgroups, South and Central America had the highest prevalence and Africa the lowest. The factors explaining the heterogeneity were the presence of chronic kidney disease, diagnostic method for peripheral arterial disease, and quality. The estimates presented very low certainty of evidence.

### Conclusions

The overall prevalence of diabetic foot at risk is high. The high heterogeneity between continents can be explained by methodological aspects and the type of population. However,

**Data Availability Statement:** All relevant data are within the manuscript and its Supporting Information files.

**Funding:** The authors received no specific funding for this work.

**Competing interests:** The authors have declared that no competing interests exist.

using the same classification is necessary for standardization of the way of measuring the components, as well as better designed general population-based studies.

## Introduction

Globally, one out of every six patients with diabetes mellitus (DM) will develop an ulcer in the lower limb at some point in their lives. Likewise, 85% of the patients with lower limb amputation, presented an ulcer that could be prevented [1]. In the US, one third of the direct costs of diabetes are caused by diabetic foot problems. The investment in diabetic foot treatment is similar to the expense related to oncological diseases [2]. The high disease burden requires the installment of an early screening to decrease these outcomes.

The main risk factors for diabetic foot ulcer are peripheral neuropathy (PN), peripheral artery disease (PAD), biomechanical deformity and limited articular mobility. Diverse academic organizations established the "foot at risk" terminology, in presence of any of them or combination [3]. There are several classification systems, sharing similar categories and components. One of the most frequently used is the International Working Group on Diabetic Foot (IWGDF) classification [4]. It allows a practical and accessible evaluation in limited-resource environments. Furthermore, it has already been assessed by validation studies in prospective studies [5]. Therefore, the early detection of diabetic foot at risk, as well as an adequate follow-up and preventive measures are essential necessities [6].

According to a global systematic review of PN on patients with DM, there is a broad range of prevalence's that indirectly reflect the impact of an adequate glycemia or metabolic control on the strategies from world-wide healthcare systems in the prevention of non-transmissible diseases [7]. Outlining that developing countries are overcoming an epidemiological transition, simultaneously coexisting with infectious diseases burden [8]. Given that, we aim to estimate the global prevalence of diabetic foot at risk base on the IWGDF stratification and explain the heterogeneity of the estimated proportions through a systematic review with metanalysis.

## Materials and methods

This study was conducted following the "Preferred Reporting Items for Systematic reviews and Meta-Analyses" (PRISMA) guidelines [9]. This protocol has been registered on PROSPERO database with the following code: CRD42021254275.

### Literature search

The search was conducted on Medline/PubMed, Web of Science, Scopus and Embase. The temporality of the search was restricted from 2000 until the last systematic search on March 4th of 2022, given that IWGDF stratification was created in 1999.

We used the Medical Subject Headings (MeSH) terms "prevalence", "risk", "Mass screening", "Cross-Sectional Studies", "diabetic foot", "foot ulcer", "risk assessment". As well as free terms as "screen*", "cross-sectional", "prevalenc*", "transvers*", "risk", "at-risk", "diabetic foot", "foot ulcer", "diabetic feet", "International Working Group on the Diabetic Foot", "IWGDF", "classificat*" and "stratificat*". The search strategy was modified according to the compatibility of each database, and the precision of the results was verified with previous studies.

A filter for transversal studies was included with the objective to discard other types of studies. Moreover, a manual search of grey literature was performed. The bibliographic references of all the included studies were reviewed in order to identify other studies that fulfilled the eligibility criteria. The complete search strategy can be seen at **S1 Table**.

## Eligibility criteria

The inclusion criteria were the following: (a) Studies that describe subjects with type 1 and/or type 2 DM (b) Diabetic foot at risk assessment according to IWGDF classification (c) Transversal or cohort studies (d) Any language. Were excluded: (a) Studies that included patients with active foot ulcer, who were not possible to exclude them (b) Studies with the same reference population (c) Other study designs: Case-control studies, letter to the editor and reviews.

## Selection and extraction of data

All the phases of the data selection were performed using the Rayyan software [10]. Duplicated documents were eliminated with the Brammer method on EndNote X9 [11]. Two reviewers (TMV and LFPM) independently screened the studies by tittle and abstract. Any discrepancy was solved by a third author (MYA). Finally, the same reviewers independently assessed the full text of the study to check if they fulfilled the eligibility criteria. The disagreements were solved by a third author (MYA). The complete list of included studies can be seen at **Table 1**. The excluded studies on the full text phase are presented on the **S2 Table**.

Data extraction was independently done by two reviewers (TMV and LFPM), who collected all the variables of interest on an Excel 365 spreadsheet. The data collected from each study were author, year of publication, country, study design, type of population, total population, type of healthcare center, continent, type of DM, history of chronic kidney disease (CKD). All the extracted was coded for the posterior analysis.

Likewise, the reviewers extracted the mean and standard deviation of the age and time of DM, the number of subjects according to sex, foot at risk, PN, PAD, biomechanical deformity, ulcer history, as well as the grade 0, 1, 2 and 3, the diagnostic tests used for PN, PAD and biomechanical deformity on each study.

## Risk of bias assessment

The quality of studies was assessed using the questionnaire developed by Loney [12], which evaluates the quality of the study assigning one point to each of the eight criteria grouped in 3 dimensions: validity, results and applicability. The risk of bias assessment was done by two reviewers (TMV and LFPM). All the disagreements were solved through discussion. If it was not possible to achieve consensus after an exhaustive discussion between the reviewers, the opinion of a third reviewer was considered (MYA). The majority consensus made the final decision. For this analysis, a score of seven to eight accounted for a high quality; five to six, accounted for moderate quality; three to four accounted for low quality; and two or less for very low quality. Fort the adequate sample size item, we considered a size of 373 subjects. This was calculated considering a prevalence of 41.36% reported by Lavery with a precision of 5% and a confidence level of 95% [13]. A detailed analysis of the evaluation can be seen in **S3 Table**.

## Statistical analysis

We calculated the pooled foot at risk prevalence with the corresponding 95% confidence interval (CI), expressed as the percentage of diabetic subjects. The extracted data were quantitatively synthesized using metanalysis techniques. A Freeman-Turkey Double Arcsine

**Table 1. Characteristics of 36 included studies of Prevalence of foot at risk of ulceration according to IWGDF stratification.**

| | Author (year) | Country | Design of study | Population, age, male (%), number of centers | Diabetes: type and time of disease | Sample size | Sum of foot at risk | G1/ G2/ G3 | Peripheral neuropathy | Peripheral arterial disease | Biomechanical deformity | History of ulcer | Quality (total score) |
|---|---|---|---|---|---|---|---|---|---|---|---|---|---|
| 1 | Peters (2001) | United states | Cohort | **Population**: Hospital **Age**: 52.6 ± 10.4 **Male n (%)**: 99 (46.4) **Center**: Unicenter | **Type**: 2 **Time**: 11 ± 9.3 | 213 | 134 | 21/ 51/ 62 | 73 | 67 | 105 | 62 | Moderate (5 points) |
| 2 | Malgrange (2003) | France | Cross-sectional | **Population**: Hospital **Age**: 56 ± 15 **Male n (%)**: 286 (51.3) **Center**: Unicenter | **Type**: 1 and 2 **Time**: 13 ± 10.4 | 555 | 151 | 54/ 54/ 43 | 150 | 94 | 117 | 35 | Moderate (6 points) |
| 3 | Mugambi (2009) | Kenia | Cross-sectional | **Population**: Hospital **Age**: 55.9 ± 9.8 **Male n (%)**: 120 (55.0) **Center**: Unicenter | **Type**: 1 y 2 **Time**: 13.8 ± 5.4 | 218 | 94 | 22/ 35/ 37 | 92 | 26 | 98 | 37 | Moderate (5 points) |
| 4 | Gonzalez de la Torre (2010) | Spain | Cross-sectional | **Population**: Primary **Age**: 64.35 ± 12.22 **Male n (%)**: 54 (56.2) **Center**: Unicenter | **Type**: 1 y 2 **Time**: - | 96 | 53 | 14/ 25/ 14 | - | 16 | 18 | 14 | Moderate (5 points) |
| 5 | Monteiro-Soares (2012) | Portugal | Cohort | **Population**: Hospital **Age**: 65 ± 10.6 **Male n (%)**: 177 (48.6) **Center**: Multicenter | **Type**: 2 **Time**: 17 ± 10.7 | 364 | 237 | 21/ 90/ 126 | 183 | 46 | 259 | 128 | Moderate (5 points) |
| 6 | Shahbazian (2013) | Iran | Cross-sectional | **Population**: Hospital **Age**: 53.8 ± 10.7 **Male n (%)**: 161 (37.4) **Center**: Unicenter | **Type**: 1 y 2 **Time**: 8.1 ± 6.6 | 430 | 153 | 75/ 47/ 31 | 264 | 7 | 81 | 31 | Moderate (5 points) |
| 7 | Bortoletto MS (2014) | Brazil | Cross-sectional | **Population**: Primaria **Age**: ND **Male n (%)**: 136 (40.3) **Center**: Multicenter | **Type**: 1 y 2 **Time**: ND | 337 | 97 | 3/ 82/ 12 | - | - | - | 12 | Moderate (5 points) |
| 8 | Alonso-Fernández (2014) | Spain | Cross-sectional | **Population**: Primaria **Age**: 68.9 ± 12 **Male n (%)**: 213 (48.0) **Center**: Multicenter | **Type**: 2 **Time**: 9.2 ± 6.4 | 443 | 93 | 53/ 23/ 17 | 38 | 42 | 64 | 17 | Moderate (6 points) |

(*Continued*)

**Table 1.** (Continued)

| | Author (year) | Country | Design of study | Population, age, male (%), number of centers | Diabetes: type and time of disease | Sample size | Sum of foot at risk | G1/ G2/ G3 | Peripheral neuropathy | Peripheral arterial disease | Biomechanical deformity | History of ulcer | Quality (total score) |
|---|---|---|---|---|---|---|---|---|---|---|---|---|---|
| 9 | Tshitenge (2014) | Botswana | Cross-sectional | **Population:** Hospital **Age:** ND **Male n (%):** 46 (31.9) **Center:** Unicenter | **Type:** 2 **Time:** ND | 144 | 22 | 10/ 7/5 | - | - | - | 5 | Moderate (5 points) |
| 10 | Wu (2015) | China | Cross-sectional | **Population:** Hospital **Age:** 59.77 ± 11.83 **Male n (%):** 156 (52.7) **Center:** Unicenter | **Type:** 1 y 2 **Time:** 7.4 ± 5.78 | 296 | 192 | 47/ 108/ 37 | 196 | - | 80 | 37 | Moderate (5 points) |
| 11 | Isip, (2016) | Philippines | Cross-sectional | **Population:** Hospital **Age:** 63.8 ± 9.42 **Male n (%):** 46 (27.0) **Center:** Unicenter | **Type:** 1 y 2 **Time:** - | 170 | 107 | 29/ 62/ 16 | 97 | 32 | 53 | 16 | Moderate (5 points) |
| 12 | Damas-Casani (2017) | Peru | Cross-sectional | **Population:** Hospital **Age:** 60.3 ±11.1 **Male n (%):** 96 (25.9) **Center:** Unicentro | **Type:** 2 **Time:** - | 370 | 288 | 29/ 235/ 24 | 131 | 145 | 201 | 24 | Moderate (5 points) |
| 13 | Kahn (2017) | Pakistan | Cross-sectional | **Population:** Primary **Age:** 53.82 ±9.96 **Male n (%):** - **Center:** Unicentro | **Type:** 2 **Time:** 7.87 ± 5.50 | 230 | 75 | 37/ 6/32 | 70 | 30 | 11 | 32 | Moderate (5 points) |
| 14 | Lucoveis (2018) | Brazil | Cross-sectional | **Population:** Hospital **Age:** ND **Male n (%):** 18 (36.0) **Center:** Unicenter | **Type:** 1 y 2 **Time:** ND | 50 | 17 | 8/3/ 6 | - | 6 | 11 | 6 | Moderate (5 points) |
| 15 | Rodríguez (2018) | Peru | Cross-sectional | **Population:** Primary **Age:** ND **Male n (%):** 122 (40.5) **Center:** Unicenter | **Type:** 2 **Time:** ND | 301 | 40 | 12/ 28/0 | 40 | 56 | 193 | - | Moderate (5 points) |
| 16 | Vibha (2018) | India | Cross-sectional | **Population:** Hospital **Age:** 63.37 ±10.8 **Male n (%):** 264 (42.5) **Center:** Unicenter | **Type:** 2 **Time:** ND | 620 | 321 | 194/ 74/ 53 | 321 | 67 | 65 | 53 | Moderate (6 points) |

(*Continued*)

**Table 1.** (Continued)

| | Author (year) | Country | Design of study | Population, age, male (%), number of centers | Diabetes: type and time of disease | Sample size | Sum of foot at risk | G1/ G2/ G3 | Peripheral neuropathy | Peripheral arterial disease | Biomechanical deformity | History of ulcer | Quality (total score) |
|---|---|---|---|---|---|---|---|---|---|---|---|---|---|
| 17 | Cardona (2018) | Cuba | Cross-sectional | **Population:** Primary **Age:** ND **Male n (%):** 145 (27.1) **Center:** Unicenter | **Type:** 2 **Time:** 11.88 ± 10.29 | 534 | 424 | 165/ 241/ 18 | 375 | 239 | 339 | 18 | Moderate (5 points) |
| 18 | Tindong (2018) | Cameroon | Cross-sectional | **Population:** Hospital **Age:** ND **Male n (%):** - **Center:** Multicenter | **Type:** 2 **Time:** ND | 203 | 39 | 14/ 16/9 | 34 | 23 | 14 | 9 | Moderate (5 points) |
| 19 | Ramírez (2019) | Venezuela | Cross-sectional | **Population:** - **Age:** 63 ± 11 **Male n (%):** 36 (36) **Center:** Unicenter | **Type:** 1 y 2 **Time:** 13.6 ± 11.0 | 100 | 95 | 5/ 74/ 16 | - | - | - | 16 | Low (4 points) |
| 20 | Cardoso (2019) | Brazil | Cross-sectional | **Population:** Primary **Age:** 59.6±12.8 **Male n (%):** 30 (35.2) **Center:** Unicenter | **Type:** 1 y 2 **Time:** 14.5 ±9.0 | 85 | 52 | 25/ 20/7 | 50 | 25 | 5 | 7 | Moderate (5 points) |
| 21 | Banik (2020) | Bangladesh | Cross-sectional | **Population:** Primary **Age:** 51.6 ± 11.9 **Male n (%):** 445 (37.0) **Center:** Unicenter | **Type:** 2 **Time:** 6.9 ±5.9 | 1200 | 534 | 50/ 142/ 342 | - | - | - | 342 | Moderate (6 points) |
| 22 | Zantour (2020) | Tunisia | Cross-sectional | **Population:** Hospital **Age:** 55.07 ± 13.54 **Male n (%):** 106 (48.1) **Center:** Unicenter | **Type:** 1 y 2 **Time:** 9.7 ± 5.19 | 220 | 60 | 13/ 39/8 | 52 | 81 | 96 | 8 | Moderate (5 points) |
| 23 | Gonzales de la Torre (2020) | Spain | Cross-sectional | **Population:** Primary **Age:** 66.93±10.85 **Male n (%):** 102 (56.0) **Center:** Unicenter | **Type:** 1 y 2 **Time:** ND | 182 | 60 | 27/ 14/ 19 | 40 | 39 | 4 | 19 | Moderate (5 points) |
| 24 | Mizouri (2021) | Tunisia | Cross-sectional | **Population:** Hospital **Age:** 55.8 ± 14.22 **Male n (%):** 33 (40.24) **Center:** Unicenter | **Type:** 1 y 2 **Time** 9.98 ± 8.11 | 82 | 47 | 18/ 26/3 | - | - | - | 3 | Moderate (5 points) |

(*Continued*)

**Table 1.** (*Continued*)

| | Author (year) | Country | Design of study | Population, age, male (%), number of centers | Diabetes: type and time of disease | Sample size | Sum of foot at risk | G1/ G2/ G3 | Peripheral neuropathy | Peripheral arterial disease | Biomechanical deformity | History of ulcer | Quality (total score) |
|---|---|---|---|---|---|---|---|---|---|---|---|---|---|
| 25 | Castañeira (2018) | Cuba | Cross-sectional | **Population:** Hospital **Age:** - **Male n (%):** - **Center:** Unicenter | **Type:** 1 y 2 **Time:** - | 111 | 105 | 27/ 17/ 61 | - | - | - | 61 | Low (4 points) |
| 26 | Mineoka (2022) | Japan | Cross-sectional | **Population:** Hospital **Age:** - **Male n (%):** - **Center:** Unicenter | **Type:** 2 **Time:** - | 469 | 200 | 150/ 38/ 12 | - | - | - | 12 | Moderate (6 points) |
| 27 | Formiga (2020) | Brazil | Cross-sectional | **Population:** Primary **Age:** 73.3 ± 7.8 **Male n (%):** 72 (28.3) **Center:** Multicenter | **Type:** 2 **Time:** 10.1 ± 8 | 254 | 163 | 111/ 12/ 40 | - | - | - | 40 | Low (4 points) |
| 28 | Elsharawy (2012) | Saudi Arabia | Cross-sectional | **Population:** Hospital **Age:** 56.9 ± 6.27 **Male n (%):** 138 (43.4) **Center:** Unicenter | **Type:** 1 y 2 **Time:** 12.4 ± 4.2 | 318 | 144 | 88/ 43/ 13 | - | - | - | 13 | Moderate (5 points) |
| 29 | Álvarez (2015) | Cuba | Cross-sectional | **Population:** Hospital **Age:** 51 **Male n (%):** 123 (58.0) **Center:** Unicenter | **Type:** 1 y 2 **Time:** 9.88 ± 9.98 | 212 | 140 | 39/ 52/ 49 | 135 | 49 | 62 | 49 | Moderate (5 points) |
| 30 | Lavery (2003) | United States | Cohort | **Population:** Primary **Age:** 69.1 ± 11.1 **Male n (%):** 838 (50.3) **Center:** Unicenter | **Type:** 1 y 2 **Time:** 11.2 ± 9.5 | 1666 | 689 | 98/ 411/ 180 | 690 | 205 | 1051 | 180 | Moderate (6 points) |
| 31 | Ndip (2010) | London | Cross-sectional | **Population:** Hospital **Age:** 64 **Male n (%):** - **Center:** Unicenter | **Type:** 1 y 2 **Time:** - | 326 | 286 | 47/ 134/ 105 | 232 | 169 | 93 | 105 | Moderate (5 points) |
| 32 | Ndip (2010) | London United States | Cross-sectional | **Population:** Hospital **Age:** 60 ± 13 **Male n (%):** 247 (53.0) **Center:** Multicenter | **Type:** 1 y 2 **Time:** 20 ± 10 | 466 | 445 | 32/ 232/ 181 | 373 | 266 | - | 181 | Moderate (6 points) |
| 33 | Yusuf (2016) | Indonesia | Cross-sectional | **Population:** Hospital **Age:** - **Male n (%):** 94 (42.9) **Center:** Unicenter | **Type:** 2 **Time:** - | 219 | 129 | 14/ 98/ 17 | - | - | - | 17 | Moderate (5 points) |

(*Continued*)

**Table 1.** (Continued)

| | Author (year) | Country | Design of study | Population, age, male (%), number of centers | Diabetes: type and time of disease | Sample size | Sum of foot at risk | G1/G2/G3 | Peripheral neuropathy | Peripheral arterial disease | Biomechanical deformity | History of ulcer | Quality (total score) |
|---|---|---|---|---|---|---|---|---|---|---|---|---|---|
| 34 | Dòria (2016) | Spain | Cross-sectional | **Population:** Hospital **Age:** - **Male n (%):** 53 (63.8) **Center:** Multicenter | **Type:** 1 y 2 **Time:** - | 83 | 80 | 18/32/30 | - | - | - | 30 | Moderate (5 points) |
| 35 | Bañuelos (2013) | Mexico | Cross-sectional | **Population:** Primary **Age:** 58,8±12,2 **Male n (%):** 26 (29.8) **Center:** Unicenter | **Type:** 2 **Time:** 9,1 ±7,4 | 87 | 56 | 44/10/2 | 21 | 10 | 44 | 2 | Low (4 points) |
| 36 | Akila (2021) | India | Cross-sectional | **Population:** Hospital **Age:** - **Male n (%):** 85 (43.3) **Center:** Unicenter | **Type:** 1 y 2 **Time:** - | 196 | 35 | 34/1/0 | - | - | 10 | - | Moderate (5 points) |

G1: Grade 1. G2: Grade 2. G3: Grade 3.

transformation was used to stabilize the variances before applying the metanalysis [14]. Due to the high expected heterogeneity, a random effects model was employed according to the Der-Simonian and Laid method [15]. The assessment of the heterogeneity between studies was performed through a Cochrane chi-square and using the $I^2$ statistic to classify the heterogeneity degree (Low: <40%, Moderate: 30–60%. Substantial: 50–90%. High: 75–100%) [16].

To evaluate the sources of heterogeneity between primary studies, a subgroup analysis by continent, sex, type of DM, type of population, age group, time of DM and history of CKD was performed. We selected these factors due to their evidence from systematic reviews. The prevalence of foot at risk varies according to the prevalence of diabetes in the continent, demographic characteristics and presence of complications [3]. Additionally, a sensitivity analysis was run according to the methodological characteristics such as year of publication, type of study, number of centers, sample size and the quality score.

A meta regression was run following the Thompson and Higgins [17] recommendations to adjust the influence of potential confounding factors on the prevalence of diabetic foot at ulceration risk. The hypothesis that factors like age, sex, time of DM, continent, type of population, type of DM, history of CKD, year of publication, type of study, number of centers, sample $\geq$ 373, quality score, diagnostic method of PN, PAD and biomechanical deformity would influence on the estimation of the prevalence, was established. All the statistical analysis were run using STATA 15.1.

## Publication bias

The publication bias in prevalence systematic reviews implies assessing the bias of small studies. Given that, there isn't a value that predisposes to the publication of itself, also called, bias due to positive results. The funnel plot was graphed using the effect size and the standard error from the effect size. An Egger's test was performed, considering a p-value < 0.1 to determine

the asymmetry. If the former analysis demonstrates asymmetry between included studies, the trim and fill random effects model was applied. This command imputes studies and calculates a new prevalence estimate.

### Geographical assessment

We made a global representation of the prevalence of foot at risk grouped by continent, detailing the number of studies, pooled prevalence and sample size.

### Evidence certainty assessment

We assessed the certainty of the diabetic foot at risk prevalence using The grading of recommendation, assessment, development, and evaluation (GRADE) approach [18]. We based our assessment on five domains, as indicated in the GRADE manual: the study limitations (risk of bias from the studies included), imprecision (sample size and CI), indirectness (generalizability), inconsistency (heterogeneity) and publication bias [16]. The prevalence estimates evaluation was adapted. The potential result was characterized as high, moderate, low or very low. The results were summarized in a "Summary of findings table (SoF)", manually adapted from the online tool GRADE [16].

### Ethics

This research did not include people, we only evaluated published studies. It was not necessary to require participan consent. We obtained authorization from the Institutional Ethics and Research Committee of thethrough the certificate 122-CIEI-CIENTIFICA-2021.

## Results

### Search results

During the initial systematic review, a total of 739 studies were identified, from which 299 duplicates were removed. During the screening by tittle and abstract, the potentially eligible studies quantity was reduced from 440 to 47. Posteriorly, in the full-text evaluation, 18 documents were excluded (**S2 Table**). In the manual search 7 documents were added. Finally, 36 studies were included in the analysis (**Fig 1**).

### Studies characteristics

The included studies showed publication dates from 2001 to 2022. Ten studies were from Asia [19–28] and South America/Caribe [29–38], seven from Europe [39–45], five from Africa [46–50] and three from North America [13, 51, 52]. The study from Ndip et al. was conducted in the United Kingdom and United States [53]. (**Fig 2**). The most frequent population scenario was inpatient with 23 studies [19–24, 26, 27, 30, 34, 36, 37, 39, 41, 44–51, 53]. Regarding the type of DM, 23 studies [13, 19, 21, 22, 24, 26, 27, 30, 32, 33, 36–40, 43–46, 48–50, 53] didn't specify the type of DM. Only three studies included population with CKD [44, 45, 53]. According to study design, the mayority was cross-sectional [19–40, 42–50, 52, 53]. and only 7 were multicenter studies [32, 33, 35, 39, 42, 45, 48, 53]. **Table 1** shows all the descriptive information of each study, including the male population, number of centers and so on.

### Risk of bias assessment on the included studies

The quality score of the included studies varied between 4 and 6. Four studies [33, 35, 37, 52] presented a low quality, and 32 studies [13, 19–32, 34, 36, 38–51, 53, 54] a moderate quality.

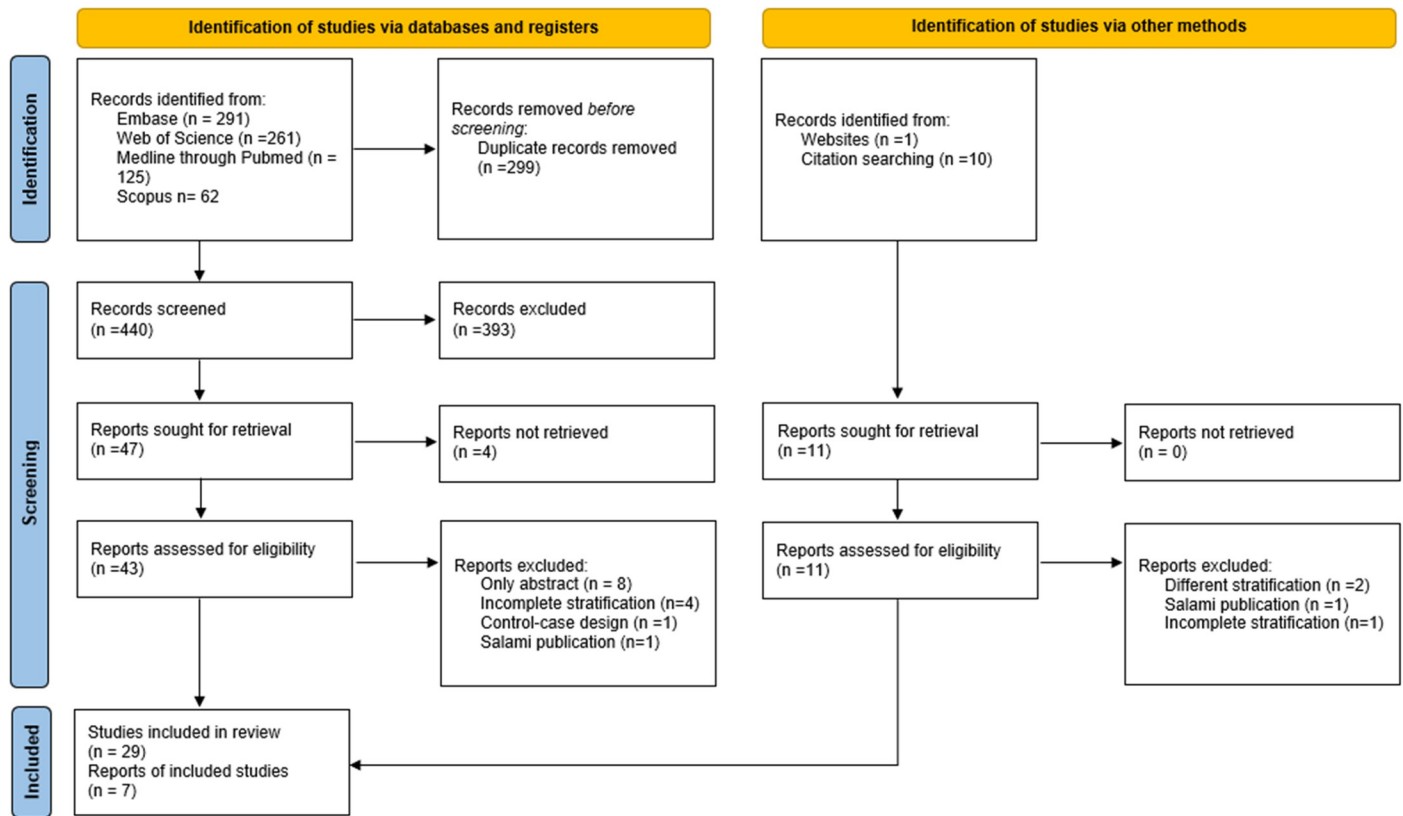

**Fig 1. Flow chart for the selection of included studies.** From: Page MJ, McKenzie JE, Bossuyt PM, Boutron I, Hoffmann TC, Mulrow CD, et al. The PRISMA 2020 statement: an updated guideline for reporting systematic reviews. BMJ 2021;372:n71. doi: 10.1136/bmj.n71. The PRISMA Statement and the PRISMA Explanation and Elaboration document are distributed under the terms of the Creative Commons Attribution License, which permits unrestricted use, distribution, and reproduction in any medium, provided the original author and source are credited. (http://prisma-statement.org/PRISMAStatement/CitingAndUsingPRISMA.aspx.

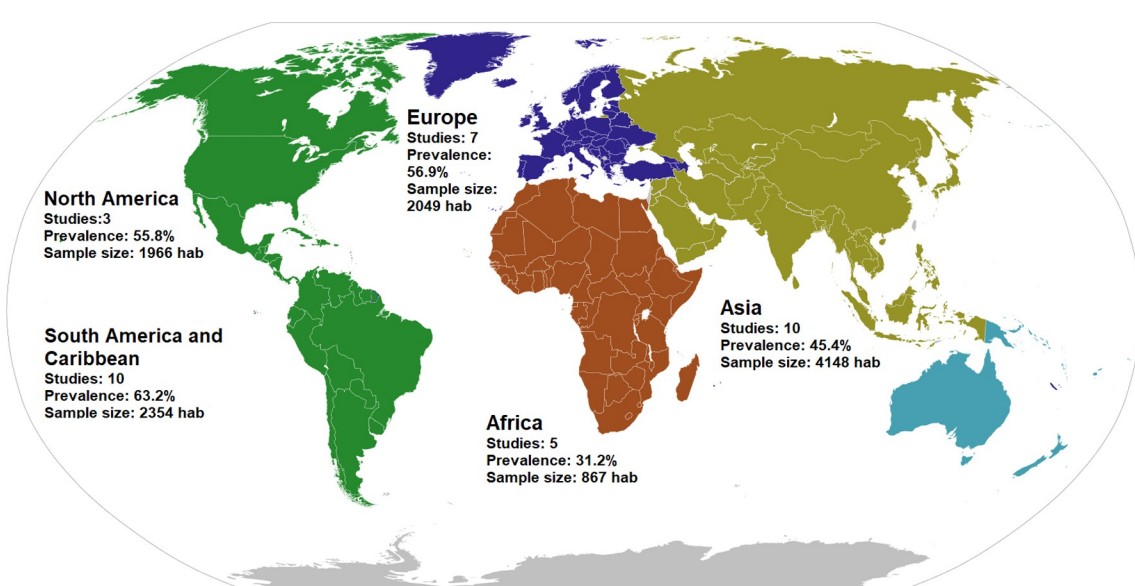

**Fig 2. Prevalence of foot at risk according to IWGDF stratification: characteristics and geographical location by continents of the included studies.** "Continental organizations" by Sbb1413 is licenced under CC-BY 4.0/Modified from original.

The most frequent unreached criteria were an inadequate sampling design (100%), inadequate sampling frame (100%), followed by an inadequate sample size (75%), and finally, not running a subgroup analysis (16%). All the studies presented validated criteria while using the IWGDF classification, according to the guideline standards, and adequately describing its population. The details of the evaluation can be observed at the **S3 Table.**

### Pooled estimations and cumulative metanalysis

The pooled prevalence of diabetic foot at risk was 53.2% (95% CI: 45.1–61.3) in 36 studies (n = 11 850). A high heterogeneity was identified ($I^2$ = 98.7%, p <0.001) (**Fig 3**).

The pooled prevalence didn't show any defined temporal variation. The estimation of the first study in 2001 was 62.9%% (95% CI: 49.5–76.3) and in 2022 was 52.0% (95% CI: 44.6–59.3). (**Fig 4**)

The most frequent components were diabetic neuropathy (42.5%; IC 95%: 42.5–51.7) and biomechanical deformity (28.9%; 95%CI 16.6–39.3).. For those who presented foot at risk, the highest percentage corresponds to grade 2 (20.2%; 95%CI 15.2–25.6). (**Table 2**).

### Subgroup analysis

In the analysis according to continent, South America and Central America (SACA) presented the highest prevalence (63.2%; 95% CI 43.9–80.6); the lowest prevalence was identified in Africa (31.3%; 95%CI 18.8–45.3).

Of the demographic variables, the highest prevalence of foot at risk was observed in those with an average age equal to or greater than 60 years. (67.0%; 95%CI 53.0–79.7) and those in which men were more than 50% (60.4%; 95%CI 41.4–78.0).

Studies carried out in hospitals or reference centers (55.6%; 95% CI 44.5–66.5), had a higher prevalence than those carried out in primary care centers (44.3%; 95% CI 33.4–55.4).

Only 13 studies specified the type of diabetes they included. The prevalence in type 2 DM was 48.9% (95%CI: 36.3–61.6). There were no studies indicating to include subjects with tipe 1 DM. In studies whose average time of diabetes was greater than 10 years, the prevalence was 66.1% CI95% 46.8–83. Population-based studies with chronic kidney disease had a prevalence of 93.4% (95% CI 87.1–97.8) (**Table 3**).

### Sensitivity analysis

The pooled prevalence, by individually excluding each study, varied between 50.60% (95%CI: 43.71–57.50) and 53.12% (95%CI: 45.89–60.42). No single study influenced on the pooled evidence estimation was identified (**Fig 5**). The published studies between 1999 and 2010 presented a prevalence of 61.2% (95%CI: 37.9–82.0) and the published between 2011 and 2022 presented a prevalence of 51.2% (42.7–59.7).

The prevalence of foot at risk was higher on low-quality studies 82.0% (95%CI: 61.3–96.0), while lower to moderate quality studies 49.3% (95%CI: 45.1–57.7). We defined an adequate sample size of a prevalence study if it was greater than 373 considering an expected frequency of 42% and a precision of 5%. According to this limit, in studies with a sample size equal or greater than 373, the prevalence was 49.9%; 95%CI 34.8–64.4). (**Table 4**).

### Metaregression

The univariate meta regression models showed an association between foot at risk prevalence with age, former history of CKD, and quality of the study (p<0.05). And a marginally significant association with time of diabetes mellitus and diagnostic method employed for PN and

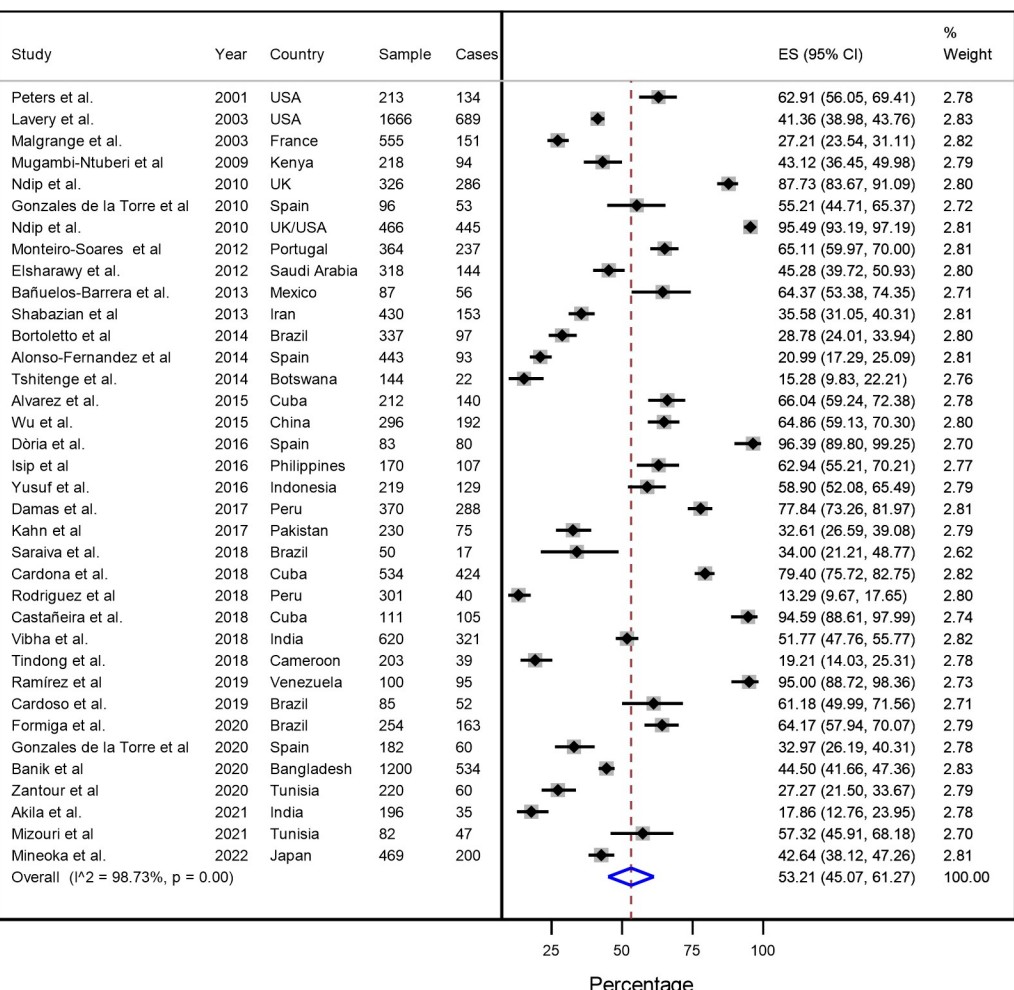

**Fig 3. Forest Plot (random effects model) of the diabetic foot at risk of ulceration metanalysis according to IWGDF stratification.**

PAD (p<0.2). In the adjusted model with these six variables, it was associated with quality of the study OR 0.67 IC95% 0.53–0.85; p = 0.002 and CKD OR 1.48 IC95% 1.11–1.97; p = 0.010. (**Table 5**).

## Publication bias

The funnel plot showed a symmetrical distribution between included studies, which indicates the lack of publication bias (small studies) (**Fig 6A**) This was confirmed by Egger's test (p = 0.351). In the "trim and fill" model, no study required "linear trimming" based imputation (**Fig 6B**).

## Certainty of evidence

We classified all available evidence as very low. Thirty four out of 36 studies presented low to moderate quality of evidence, according to Loney's scale. Additionally, the category was decreased due to the high heterogeneity ($I^2 > 75\%$) in the metanalysis. (**Table 6**)

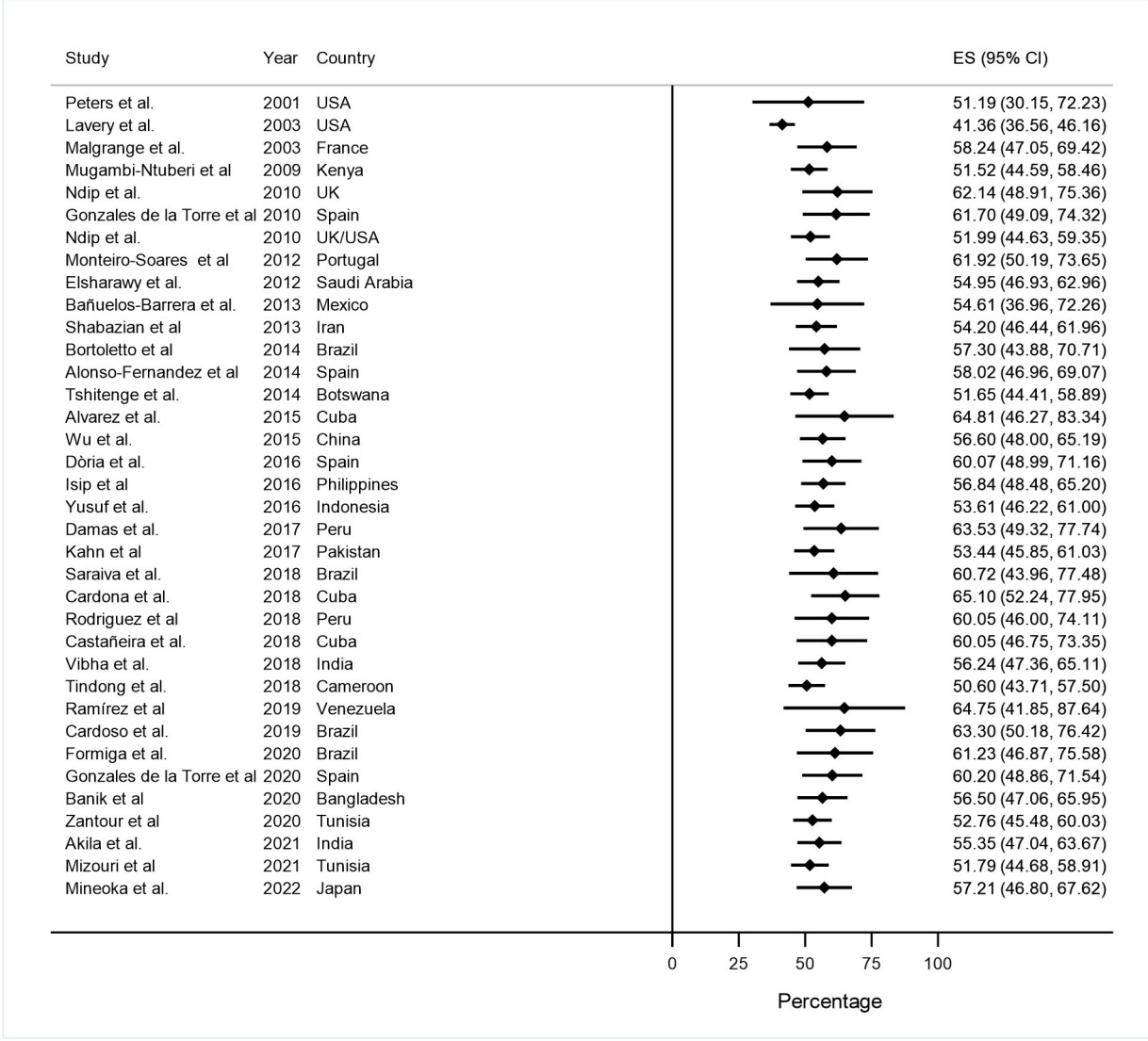

| Study | Year | Country | ES (95% CI) |
|---|---|---|---|
| Peters et al. | 2001 | USA | 51.19 (30.15, 72.23) |
| Lavery et al. | 2003 | USA | 41.36 (36.56, 46.16) |
| Malgrange et al. | 2003 | France | 58.24 (47.05, 69.42) |
| Mugambi-Ntuberi et al | 2009 | Kenya | 51.52 (44.59, 58.46) |
| Ndip et al. | 2010 | UK | 62.14 (48.91, 75.36) |
| Gonzales de la Torre et al | 2010 | Spain | 61.70 (49.09, 74.32) |
| Ndip et al. | 2010 | UK/USA | 51.99 (44.63, 59.35) |
| Monteiro-Soares et al | 2012 | Portugal | 61.92 (50.19, 73.65) |
| Elsharawy et al. | 2012 | Saudi Arabia | 54.95 (46.93, 62.96) |
| Bañuelos-Barrera et al. | 2013 | Mexico | 54.61 (36.96, 72.26) |
| Shabazian et al | 2013 | Iran | 54.20 (46.44, 61.96) |
| Bortoletto et al | 2014 | Brazil | 57.30 (43.88, 70.71) |
| Alonso-Fernandez et al | 2014 | Spain | 58.02 (46.96, 69.07) |
| Tshitenge et al. | 2014 | Botswana | 51.65 (44.41, 58.89) |
| Alvarez et al. | 2015 | Cuba | 64.81 (46.27, 83.34) |
| Wu et al. | 2015 | China | 56.60 (48.00, 65.19) |
| Dòria et al. | 2016 | Spain | 60.07 (48.99, 71.16) |
| Isip et al | 2016 | Philippines | 56.84 (48.48, 65.20) |
| Yusuf et al. | 2016 | Indonesia | 53.61 (46.22, 61.00) |
| Damas et al. | 2017 | Peru | 63.53 (49.32, 77.74) |
| Kahn et al | 2017 | Pakistan | 53.44 (45.85, 61.03) |
| Saraiva et al. | 2018 | Brazil | 60.72 (43.96, 77.48) |
| Cardona et al. | 2018 | Cuba | 65.10 (52.24, 77.95) |
| Rodriguez et al | 2018 | Peru | 60.05 (46.00, 74.11) |
| Castañeira et al. | 2018 | Cuba | 60.05 (46.75, 73.35) |
| Vibha et al. | 2018 | India | 56.24 (47.36, 65.11) |
| Tindong et al. | 2018 | Cameroon | 50.60 (43.71, 57.50) |
| Ramírez et al | 2019 | Venezuela | 64.75 (41.85, 87.64) |
| Cardoso et al. | 2019 | Brazil | 63.30 (50.18, 76.42) |
| Formiga et al. | 2020 | Brazil | 61.23 (46.87, 75.58) |
| Gonzales de la Torre et al | 2020 | Spain | 60.20 (48.86, 71.54) |
| Banik et al | 2020 | Bangladesh | 56.50 (47.06, 65.95) |
| Zantour et al | 2020 | Tunisia | 52.76 (45.48, 60.03) |
| Akila et al. | 2021 | India | 55.35 (47.04, 63.67) |
| Mizouri et al | 2021 | Tunisia | 51.79 (44.68, 58.91) |
| Mineoka et al. | 2022 | Japan | 57.21 (46.80, 67.62) |

Percentage

**Fig 4. Cumulative metanalysis of the diabetic foot at ulceration risk according to IWGDF stratification.**

## Discussion

The diabetic foot is one of the most expensive complications in diabetes mellitus. Identifying and following the patients with foot at risk is a priority to decrease the burden of this disease. In this systematic review with meta-analysis, we estimated a global foot at risk prevalence of 53.2% (CI95%: 45.0 to 61.2). No time-cumulative prevalence tendency of foot at risk of ulceration was identified. The main sources of heterogeneity related to a major prevalence were the time duration of DM and former history of chronic kidney disease (CKD). Drawing of the high heterogeneity and high risk of bias, the included evidence was considered of very low certainty.

The IWGDF scale, designed in 1999, determined risk of ulceration if at least one had peripheral neuropathy. In 2008 the classification was re-evaluated and they divided stages 2 and 3. Stage 2 in 2A (PN + deformity) and 2B (just PAD). Likewise, stage 3 in in 3A (former ulcer history) and 3B (former amputation history) [54]. In our study, five reports [28, 41, 43, 50, 51] used this last stratification. But this modification didn't change the percentage of

**Table 2. Prevalence of grades and components of diabetic foot at risk of ulceration according to IWGDF stratification.**

|  | N | Prevalence | 95% IC | I² |
|---|---|---|---|---|
| **Grade** |  |  |  |  |
| Grade 0 | 36 | 44.4% | 36.3–52.7 | 98.8% |
| Grade 1 | 36 | 14.6% | 11.1–18.4 | 96.8% |
| Grade 2 | 36 | 20.2% | 15.2–25.6 | 98.0% |
| Grade 3 | 36 | 11.3% | 7.9–15.2 | 97.4% |
| **Components** |  |  |  |  |
| Peripheral neuropathy | 22 | 42.5% | 33.5–51.70 | 98.6% |
| Peripheral artery disease | 23 | 20.8% | 14.7–27.7 | 98.0% |
| Biomechanical deformity | 24 | 28.9% | 19.6–39.3 | 99.0% |
| Previous history of ulcer | 36 | 11.3% | 7.9–15.2 | 97.4% |
| **Diagnostic method of peripheral neuropathy** |  |  |  |  |
| Monofilament or diapason or other diagnostic method | 11 | 44.6% | 29.3–60.5 | 98.3% |
| Two objective tests | 9 | 51.9% | 39.4–64.2 | 98.2% |
| Three or more tests | 16 | 59.8% | 46.9–72.1 | 98.7% |
| **Diagnostic method or peripheral artery disease** |  |  |  |  |
| Pulses palpation or former history | 14 | 46.0% | 30.9–61.4 | 98.8% |
| Ankle-Brachial Index | 17 | 53.7% | 45.6–61.6 | 97.5% |
| Ankle-Brachial Index and another method | 5 | 70.8% | 43.2–92.0 | 99.3% |

patients at risk. In 2019, the classification presents another modification. The minimum requirement to be at risk was to have PN or PAD. This change will increase the frequency of standing at risk [4]. In a Latin-American serie, frequency increased from 37.4% to 54.3% with these new criteria [55].

There are multiple methods to diagnose peripheral neuropathy. The Toronto Consensus classified accuracy according to the methods used.. Possible PN when symptoms can be identified or there is an objective test. Probable PN when two objective tests were run. Confirmed PN when abnormal nervous conduction test with a sign or symptom. Subclinical PN when there is only abnormal nervous conduction test [56]. A systematic review of peripheral neuropathy, which included studies that didn't describe the measuring method, obtained a prevalence of 30% of PN (IC95% 25–34%); I2 = 99.5%; <0.001 [7]. Another SR in SACA, which specified just including studies with at least two objective tests obtained a prevalence of 46.5% IC95% 38–55%; I2 = 98.2%; p<0.001 [57]. In our systematic review, 69% of included studies employed two or more objective tests, which makes data quality to be acceptable..

For screening of peripheral arterial disease, clinical guidelines recommend pulse palpation, ankle brachial index, and arterial waveform analysis. Arterial calcification interferes with an adequate interpretation of the ABI. We obtain high ITB values (greater than 1.4). However, this is due to arteriosclerosis that causes difficulty in arterial obliteration [58]. In our systematic review, 61% of the studies performed the ABI, furthermore, five of them performed a second confirmatory test. Considering good quality data as well.

The greatest prevalence was found on SACA, followed by NA and Europe on the same percentage. While the lowest prevalence was observed in Africa. Factors that might influence are the local diabetes prevalence, proportion of subjects with diabetes who are unaware of having the disease, the incidence rate, government strategies to prevent complications, if the subjects were from centers of reference or the general population. mong other.

According to the International Diabetes Federation, Africa has the lowest prevalence of DM [59], however, it will have the highest DM percentage increase and also the highest

**Table 3. Subgroup analysis of the prevalence of diabetic foot at risk of ulceration according to IWGDF stratification meta-analysis.**

|  | N | Prevalence | 95% CI | % Weight | I² |
|---|---|---|---|---|---|
| **Continent** |  |  |  |  |  |
| North America | 3 | 55.9% | 38.4–72.6 | 8.3 | . |
| South and Central America | 10 | 63.2% | 43.9–80.6 | 27.6 | 98.9% |
| Europe | 7 | 56.9% | 33.2–79.1 | 19.4 | 99.1% |
| Asia | 10 | 45.4% | 38.1–52.8 | 28.00 | 95.5% |
| Africa | 5 | 31.3% | 18.8–45.3 | 13.8 | 94.5% |
| More than one continent | 1 | 95.5% | 45.1–61.3 | 2.8 | . |
| **Age** |  |  |  |  |  |
| < 60 years | 14 | 45.6% | 37.8–53.6 | 38.9 | 96.1% |
| ≥60 years | 14 | 67.0% | 53.0–79.7 | 39.0 | 99.0% |
| Undetermined | 8 | 41.4% | 20.8–63.8 | 22.2 | 99.1% |
| **Diabetes time** |  |  |  |  |  |
| < 10 years | 11 | 46.6% | 38.4–54.9 | 30.6 | 96.9% |
| ≥10 years | 9 | 66.1% | 46.8–83.0 | 25.1 | 99.1% |
| Undetermined | 16 | 50.2% | 36.0–64.3 | 44.3 | 98.7% |
| **Male** |  |  |  |  |  |
| <50% | 22 | 49.2% | 39.7–58.7 | 61.0 | 98.4% |
| ≥50% | 9 | 60.4% | 41.4–78.0 | 25.0 | 99.2% |
| Undetermined | 5 | 57.7% | 28.0–84.6 | 13.9 | 99.2% |
| **Type of population** |  |  |  |  |  |
| In-patient | 23 | 55.6% | 44.5–66.5 | 63.9 | 98.7% |
| Primary care | 12 | 44.3% | 33.4–55.4 | 33.4 | 98.4% |
| Any population | 1 | 95.0% | 88.7–98.4 | 2.7 | . |
| **Type of DM** |  |  |  |  |  |
| Unspecified | 23 | 55.7% | 44.7–66.4 | 63.7 | 98.8% |
| Type 2 | 13 | 48.9% | 36.3–61.6 | 36.3 | 98.7% |
| **Chronic kidney disease (CKD)** |  |  |  |  |  |
| No | 33 | 48.7% | 41.7–55.7 | 91.7 | 98.1% |
| Yes | 3 | 93.4% | 87.1–97.8 | 8.3 | . |

proportion of undiagnosed DM for 2045. Despite its low current prevalence, it can become one of the first ranked continenets in the near future. The high prevalence of SACA, NA and Europe may be due to the predominance of hospital-based studies. Where glycemic and metabolic control is often inadequate. Hospital series show that only 30 to 50% of the subjects achieve a glycated hemoglobin level of less than 7% and only one subject out of eight also controls blood pressure and lipids.. North America and Europe, mainly the USA, despite having the largest health budgets in the world, the inequity in health does not allow consistent results for the entire population.. There are no representative studies of the Western Pacific area, which have the highest prevalence in the world. We could speculate that they have much higher prevalences of foot at risk than those described [59].

Understanding the diabetic foot as a continuum, a systematic review of the presence of foot ulcers globally found an estimated prevalence of 6.3%. The highest prevalence was identified at North America with 13%, followed by Africa with 7.2%, Asia 5.5% and Europe 5.1%. The report didn't include any studies from South America, because of not finding publications in the English language. Given the high diabetic foot at risk rate on SACA, it is expected to be one of the highest in the world. Africa went from lowest standing at risk to second most likely to have cold sores, probably due to limitations in its overall health service. The high prevalence

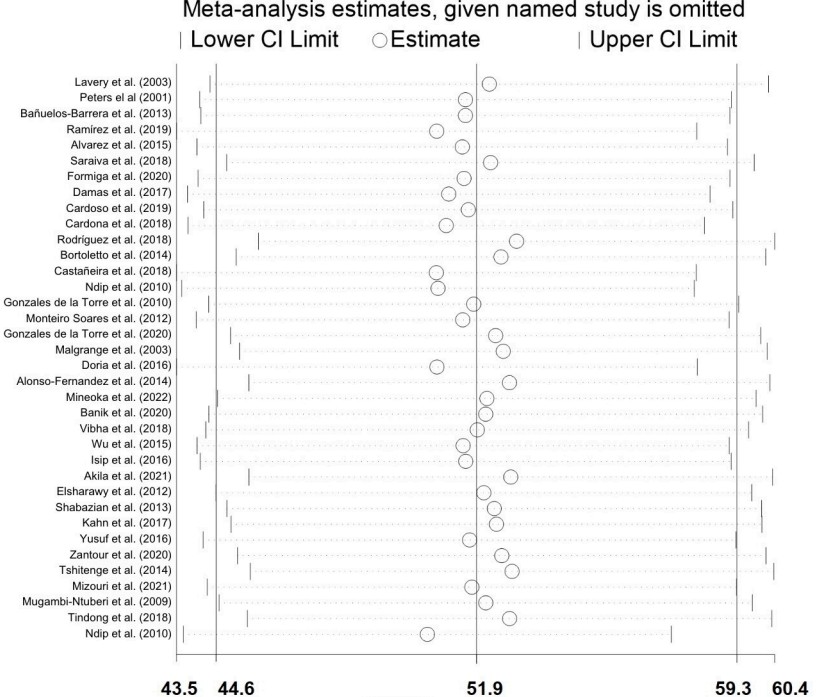

**Fig 5. Sensitivity metanalysis by the consecutive exclusion of the 36 included studies.**

of NA, with data mainly from the USA, would be made up of a population with limited access to health, despite having the health system with the largest budget [60].

A longer period of diabetes and presence of diabetic complications are associated with a higher probability of having a foot at risk. The chronic kidney disease (CKD) exacerbates the oxidative stress, inflammation, and endothelial distress [61]. Increases the occurrence of PAD by 2.5 times. and 90% of the dialysis population has peripheral neuropathy [62]., debuting as

**Table 4. Sensitivity analysis of the prevalence of diabetic foot at risk of ulceration according to IWGDF stratification meta-analysis.**

|  | N | Prevalence | 95% CI | % Weight | I² |
|---|---|---|---|---|---|
| **Year** | | | | | |
| 1999–2010 | 7 | 61.2% | 37.9–82.0 | 19.6 | 99.4% |
| 2011–2022 | 29 | 51.2% | 42.7–59.7 | 80.4 | 98.4% |
| **Study design** | | | | | |
| Transversal | 33 | 52.9% | 43.7–62.1 | 91.6 | 98.9% |
| Cohort | 3 | 56.4% | 38.6–73.4 | 8.4 | . |
| **Number of centers** | | | | | |
| Single center | 29 | 53.3% | 45.6–61.0 | 80.5 | 98.2% |
| Multicenter | 7 | 52.8% | 24.9–79.7 | 19.5 | 99.5% |
| **Sample size ≥ 373 subjects** | | | | | |
| No | 27 | 54.3% | 44.1–64.4 | 74.6 | 98.3% |
| Yes | 9 | 49.9% | 34.8–65.1 | 25.4 | 99.3% |
| **Quality score** | | | | | |
| Low | 4 | 82.0% | 61.3–96.0 | 11.0 | 96.5% |
| Moderate | 32 | 49.3% | 45.1–57.7 | 89.0 | 98.7% |

**Table 5. Meta regression model of the prevalence of diabetic foot at risk of ulceration according to IWGDF stratification.**

| | | Crude | | | Adjusted | | |
|---|---|---|---|---|---|---|---|
| | N | OR | 95% CI | p-value | OR | 95% CI | p-value |
| **Continent** | | | | | | | |
| North America | 3 | 1.00 | | | | | |
| South and Central America | 10 | 1.06 | 0.79–1.44 | 0.694 | | | |
| Europe | 7 | 0.98 | 0.72–1.35 | 0.932 | | | |
| Asia | 10 | 0.91 | 0.66–1.23 | 0.518 | | | |
| Africa | 5 | 0.79 | 0.56–1.11 | 0.170 | | | |
| More than one continent | 1 | 1.49 | 1.33–2.27 | 0.130 | | | |
| **Age** | | | | | | | |
| Mean <60 years | 14 | 1.00 | | | 1.00 | | |
| Mean ≥ 60 years | 14 | 1.20 | 1.01–1.43 | 0.041 | 1.04 | 0.85–1.27 | 0.689 |
| Undetermined | 8 | 0.95 | 0.77–1.17 | 0.642 | 0.94 | 0.74–1.19 | 0.591 |
| **Time of diabetes** | | | | | | | |
| Mean <10 years | 11 | 1.00 | | | 1.00 | | |
| Mean ≥ 10 years | 9 | 1.18 | 0.95–1.47 | 0.128 | 1.16 | 0.98–1.39 | 0.088 |
| Undetermined | 16 | 1.02 | 0.84–1.24 | 0.805 | 1.09 | 0.88–1.36 | 0.415 |
| **Male** | | | | | | | |
| < 50% of the sample | 22 | 1.00 | | | | | |
| ≥ 50% of the sample | 9 | 1.08 | 0.89–1.32 | 0.394 | | | |
| Undetermined | 5 | 1.06 | 0.83–1.36 | 0.634 | | | |
| **Type of population** | | | | | | | |
| In-patient | 23 | 1.00 | | | | | |
| Primary care | 12 | 1.10 | 0.93–1.31 | 0.248 | | | |
| Any population | 1 | 1.65 | 0.99–2.78 | 0.056 | | | |
| **Type of DM** | | | | | | | |
| Both | 23 | 1.00 | | | | | |
| Type 2 | 13 | 1.04 | 0.88–1,24 | 0.626 | | | |
| **Chronic kidney disease (CKD)** | | | | | | | |
| No | 33 | 1.00 | | | 1.00 | | |
| Yes | 3 | 1.56 | 1.21–2.03 | 0.001 | 1.48 | 1.11–1.97 | 0.010 |
| **Year** | | | | | | | |
| 1999–2010 | 7 | 1.00 | | | | | |
| 2011–2022 | 29 | 0.92 | 0.74–1.13 | 0.403 | | | |
| **Type of study** | | | | | | | |
| Transversal | 33 | 1.00 | | | | | |
| Cohort | 3 | 1.05 | 0.78–1.41 | 0.758 | | | |
| **Number of centers** | | | | | | | |
| Single center | 29 | 1.00 | | | | | |
| Multicenter | 7 | 0.97 | 0.79–1.19 | 0.763 | | | |
| **Sample size ≥ 373** | | | | | | | |
| No | 27 | 1.00 | | | | | |
| Yes | 9 | 0.97 | 0.79–1.15 | 0.634 | | | |
| **Quality score** | | | | | | | |
| Low | 4 | 1.00 | | | 1.00 | | |
| Moderate | 32 | 0.73 | 0.57–0.95 | 0.020 | 0.68 | 0.54–0.85 | 0.002 |
| **Diagnostic method of PN** | | | | | | | |
| Just one test | 11 | 1.00 | | | 1.00 | | |

(*Continued*)

**Table 5.** (Continued)

|  | | Crude | | | Adjusted | | |
|---|---|---|---|---|---|---|---|
|  | N | OR | 95% CI | p-value | OR | 95% CI | p-value |
| Two objective tests | 9 | 1.09 | 0.88–1.37 | 0.403 | 1.14 | 0.91–1.45 | 0.231 |
| Three or more tests | 16 | 1.17 | 0.96–1.42 | 0.114 | 1.15 | 0.94–1.41 | 0.155 |
| **Diagnostic method of peripheral artery disease (PAD)** | | | | | | | |
| Just pulses or former history | 14 | 1.00 | | | 1.00 | | |
| Ankle-Brachial Index | 17 | 1.08 | 0.91–1.29 | 0.350 | 1.02 | 0.84–1.22 | 0.846 |
| Ankle-Brachial Index and another method | 5 | 1.26 | 0.98–1.62 | 0.066 | 1.03 | 0.81–1.32 | 0.780 |

Adjusted by age, diabetes time, kidney disease, quality of the study, diagnostic method of PN and PAD

loss of sensation, weakness and pain [63]. For these reasons, prevalence of foot at risk of ulceration in patients with CKD is much higher than in patients without CKD (93.44% vs 48.68%). Likewise, poor control of diabetes is also a factor for the development of PN and PAD [64]. Glycemic control is lost over time, despite intensive use of secretagogues and metformin. We must avoid therapeutic inertia and start insulin or drugs with other non-insulin secretagogue action mechanisms to improve metabolic control and therefore the components of the foot at risk [65].The population has many myths about the use of insulin that we must manage properly to improve metabolic control [66].

## Impact in public health

The IWGDF criteria has been validated to establish the risk of ulcer, and this starts the downstream cascade that will end on a major amputation event. The risk determination is just the first step. It was run with limited resources or with sophisticated equipment. This must be followed by a follow-up schedule according to the identified risk, as well as the treatment of altered factors by a multidisciplinary team. Finally, the patient must achieve discipline in self-healthcare [67]. Initially harmless events like an inadequate cut foot nail, lack of foot hydration,

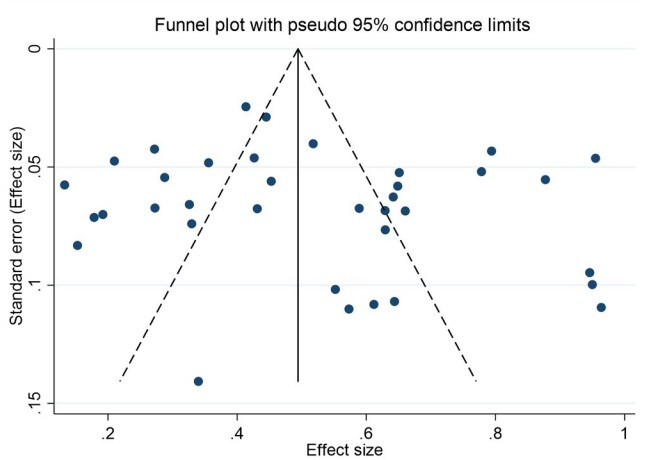
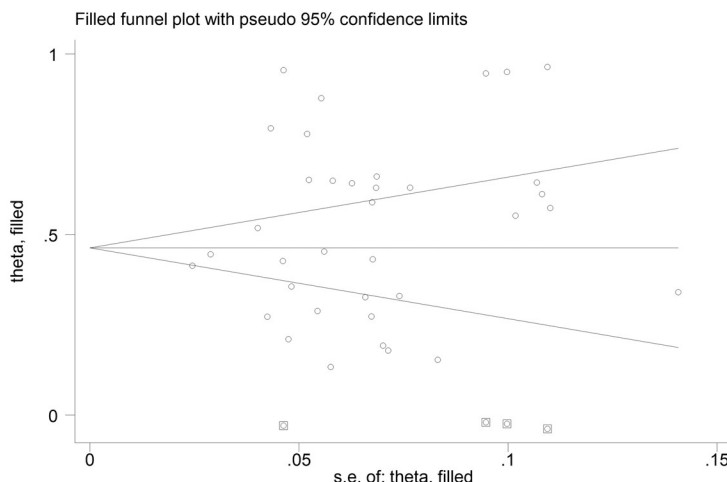

**Fig 6.** A. Diabetic foot at risk of ulceration according to IWGDF stratification of the 36 included studies Funnel Plot. A—Classic Funnel Plot. B. Diabetic foot at risk of ulceration according to IWGDF stratification of the 36 included studies Funnel Plot. B–Funnel Plot trimmed studies.

**Table 6. Quality of the body of evidence according to GRADE: Summary of findings.**

| Results | Anticipated absolute effects (95% CI) | | № of participants | Certainty of evidence |
|---|---|---|---|---|
| | Pooled frequency (%) | IC 95% | (Studies) | (GRADE) |
| Prevalence of diabetic foot at risk of ulceration according to IWGDF stratification | 53.2 | 45.1–61.3 | 11850 participants on 36 studies | ⊕◯◯◯[a,b,c,d] Very low |

CI, Confidence interval; IWGDF, International Working Group on the Diabetic Foot

a The grade of certainty starts on a low certainty because no population-based studies were included

b High risk of bias was detected (low and moderate quality according to Loney scale) in 36 studies

c High inconsistency of the metanalysis was detected. The calculated I2 was > 75%

d No imprecision due to adequate sample size and narrow confidence interval

using a compressive shoe, walking shoeless, loss of protective sensibility, will increase the likelihoods of developing an ulcer in these patients.

Evidence show that PAD is as important as PN for ulcer development. The higher life expectancy increased the prevalence of the former factors. By knowing the importance of doing this screening and the time invested in evaluating all the components, the Canadian Diabetes Society have designed and validated a reduced but complete version that can be applied in 60 seconds. This with the aim of screening the greatest number of patients possible [60]. The primary care services should be capable of providing the patients with the screening and follow-up. However, they are usually overcrowded to face this condition. Government strategies are required to face the classic infectious problems and emergent non-transmissible diseases simultaneously.

## Limitations

This study presents some limitations. Only 23 countries were included out of a total of 195 countries in the world. The worst represented continent was Africa. Many studies were excluded for not having free access, even after contacting the corresponding authors. Furthermore, the data was based on series of inpatient or primary care patients, with limited external generalization because no general population study was included. The extrapolation of this data must be performed with precaution. Despite using the same criteria for diabetic foot at risk, many authors used different forms of measuring neuropathy or peripheral artery disease. As method to maintain the quality, we only included studies that described the methods used. We outline that 70% of included studies used an adequate method for peripheral neuropathy (two or more methods) and 60% used the Ankle-Brachial Index for peripheral artery disease. At last, some studies didn't present data on sex, age, time of diabetes mellitus disease, thus, the effect of these factors might be underestimated.

The strengths were that exhaustive systematic search to incorporate all the studies, without language nor year of publication restrictions. Moreover, to decrease the measurement heterogeneity, despite existing multiple diabetic foot at risk scales [5], we used one scale with external validation and with a world-wide application. Meta-analysis techniques to assess the sources of heterogeneity were used. Finally, an exhaustive assessment of quality and certainty of evidence of the calculated estimates was performed, in order to guide the design of future studies.

## Recommendations for research

Given the multiple forms of evaluation of neuropathy. Comparing possible versus probable neuropathy according to the Toronto Consensus in ulcer development is important. Likewise,

we generally find 30% arterial calcification in the ABI that limits the determination of flow. Therefore, we must validate adding the evaluation of the arterial waveform by portable vascular Doppler for ulcer occurrence.

Carrying out studies based on the hospital population is more feasible. But we should know the frequency of foot at risk in the general population, where the prevalence is lower, and we find people with better diabetes control or less access to health services.

## Conclusion

The overall prevalence of foot at risk is high on worldwide. We identified high between-study heterogeneity and significant limitations (e.g limited countries, heterogeneous definitions, hospital-based studies, low certainty level of evidence). We need upgraded research using standardized and population-based studies and urgent action against preventable causes of diabetic foot.

## Supporting information

**S1 Checklist. Prisma checklist of items include reporting a systematic review.**
(DOCX)

**S1 Table. Search strategy.**
(DOCX)

**S2 Table. Studies that were evaluated in full-text and were excluded.**
(DOCX)

**S3 Table. Evaluation of the quality of prevalence studies.**
(DOCX)

## Acknowledgments

We would like to thank to Dirección General de Investigación, Desarrollo e Innovación of the Universidad Científica del Sur for their assistance with the logistic aspects of this study.

## Author Contributions

**Conceptualization:** Tania Maldonado-Valer, Luis F. Pareja-Mujica, Marlon Yovera-Aldana.

**Data curation:** Fernando Andres Terry-Escalante, Mylenka Jennifer Chevarría-Arriaga, Marlon Yovera-Aldana.

**Formal analysis:** Rodrigo Corcuera-Ciudad, Marlon Yovera-Aldana.

**Investigation:** Tania Maldonado-Valer, Luis F. Pareja-Mujica, Rodrigo Corcuera-Ciudad, Fernando Andres Terry-Escalante, Mylenka Jennifer Chevarría-Arriaga, Marlon Yovera-Aldana.

**Methodology:** Tania Maldonado-Valer, Marlon Yovera-Aldana.

**Project administration:** Tania Maldonado-Valer, Marlon Yovera-Aldana.

**Resources:** Tania Maldonado-Valer, Luis F. Pareja-Mujica, Marlon Yovera-Aldana.

**Supervision:** Tania Maldonado-Valer, Marlon Yovera-Aldana.

**Validation:** Tery Vasquez-Hassinger, Marlon Yovera-Aldana.

**Visualization:** Tery Vasquez-Hassinger, Marlon Yovera-Aldana.

**Writing – original draft:** Tania Maldonado-Valer, Luis F. Pareja-Mujica,
    Rodrigo Corcuera-Ciudad, Fernando Andres Terry-Escalante,
    Mylenka Jennifer Chevarría-Arriaga, Marlon Yovera-Aldana.

**Writing – review & editing:** Tania Maldonado-Valer, Tery Vasquez-Hassinger,
    Marlon Yovera-Aldana.

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
