## [Decision Letter · Decision Letter 0]

2 Jan 2023

PONE-D-22-29064Prevalence of diabetic foot at risk of ulcer development and its components stratification according to the International Working Group on the Diabetic Foot (IWGDF): A systematic review with metanalysis

PLOS ONE

Dear Dr. Marlon Yovera-Aldana

Thank you for submitting your manuscript to PLOS ONE. After careful consideration, we feel that it has merit but does not fully meet PLOS ONE’s publication criteria as it currently stands. Therefore, we invite you to submit a revised version of the manuscript that addresses the points raised during the review process.

ACADEMIC EDITOR :   

This is a potentially remarkable review paper looking at the global prevalence of diabetic foot at risk base on the IWGDF stratification. I do have some comments/suggestions that I feel would improve the manuscript.

introduction 

Does the first sentence refer to worldwide estimates? To any diabetes?I do not understand the sentence towards the end of para 1 " Moreover, the healthcare costs for diabetic foot treatment equals costs related to oncological fields, being a burden for developing countries " – what does that mean? please clarify with some statistical data? 

Methods

  The authors say that if there is significant heterogeneity they will conduct a subgroup analysis by continents, sex, type of study, type of DM, type of population, age group, time of DM and history of CKD was performed.  What is the a priori justification for selecting these variables as potential sources of heterogeneity?

Results

In general, I found it hard to wade through the results. Virtually all numbers presented in tables are reiterated in the text - this is unnecessary and I would recommend only the relevant results are emphasised with numbers.The data seem to show important regional differences e.g. prevalence rates seems to be very high in the South America and Central America (SACA) presented the highest prevalence with the lowest prevalence was identified on Africa compared to other regions.  These are important findings that need discussion in terms of implications in practices and research in particular in   limited resources countries in those regions.It was virtually impossible to read the forest plans

Discussion

 I request that authors expand the discussion to be more narrative and also forward-looking and also including a section on directions of future research.

Conclusions

 Weak conclusion which says little - can you say something more concrete

Please submit your revised manuscript by January 28, 2023. If you will need more time than this to complete your revisions, please reply to this message or contact the journal office at plosone@plos.org. Please include the following items when submitting your revised manuscript:A rebuttal letter that responds to each point raised by the academic editor and reviewer(s). You should upload this letter as a separate file labeled 'Response to Reviewers'.A marked-up copy of your manuscript that highlights changes made to the original version. You should upload this as a separate file labeled 'Revised Manuscript with Track Changes'.An unmarked version of your revised paper without tracked changes. You should upload this as a separate file labeled 'Manuscript'.If applicable, we recommend that you deposit your laboratory protocols in protocols.io to enhance the reproducibility of your results. Protocols.io assigns your protocol its own identifier (DOI) so that it can be cited independently in the future. For instructions see: https://journals.plos.org/plosone/s/submission-guidelines#loc-laboratory-protocols. Additionally, PLOS ONE offers an option for publishing peer-reviewed Lab Protocol articles, which describe protocols hosted on protocols.io. Read more information on sharing protocols at https://plos.org/protocols?utm_medium=editorial-email&utm_source=authorletters&utm_campaign=protocols.

We look forward to receiving your revised manuscript.

Kind regards,

Mahmoud M Werfalli, PhD

Academic Editor

PLOS ONE

Journal Requirements:

2. We note that Figure 2 in your submission contain [map/satellite] images which may be copyrighted. All PLOS content is published under the Creative Commons Attribution License (CC BY 4.0), which means that the manuscript, images, and Supporting Information files will be freely available online, and any third party is permitted to access, download, copy, distribute, and use these materials in any way, even commercially, with proper attribution. For these reasons, we cannot publish previously copyrighted maps or satellite images created using proprietary data, such as Google software (Google Maps, Street View, and Earth). For more information, see our copyright guidelines: http://journals.plos.org/plosone/s/licenses-and-copyright.

Additional Editor Comments (if provided):

This is a potentially remarkable review paper looking at the global prevalence of diabetic foot at risk base on the IWGDF stratification. I do have some comments/suggestions that I feel would improve the manuscript.

intro

1. Does the first sentence refer to worldwide estimates? To any diabetes?

2. I do not understand the sentence towards the end of para 1 " Moreover, the healthcare costs for diabetic foot treatment equals costs related to oncological fields, being a burden for developing countries " – what does that mean? please clarify with some statistical data?

Methods

The authors say that if there is significant heterogeneity they will conduct a subgroup analysis by continents, sex, type of study, type of DM, type of population, age group, time of DM and history of CKD was performed. What is the a prior justification for selecting these variables as potential sources of heterogeneity?

Results

1. In general, I found it hard to wade through the results. Virtually all numbers presented in tables are reiterated in the text - this is unnecessary and I would recommend only the relevant results are emphasised with numbers.

2. The data seem to show important regional differences e.g. prevalence rates seems to be very high in the South America and Central America (SACA) presented the highest prevalence with the lowest prevalence was identified on Africa compared to other regions. These are important findings that need discussion in terms of implications in practices and research in particular in limited resources countries in those regions.

3. It was virtually impossible to read the forest plans

discussion

I request that authors expand the discussion to be more narrative and also forward-looking and also including a section on directions of future research.

Conclusions

weak conclusion which says little - can you say something more concrete

Reviewers' comments:

Reviewer's Responses to Questions

**Comments to the Author**

1. Is the manuscript technically sound, and do the data support the conclusions?

Reviewer #1: Yes

2. Has the statistical analysis been performed appropriately and rigorously? 

Reviewer #1: Yes

3. Have the authors made all data underlying the findings in their manuscript fully available?

Reviewer #1: Yes

4. Is the manuscript presented in an intelligible fashion and written in standard English?

Reviewer #1: Yes

5. Review Comments to the Author

Reviewer #1: The authors have done a good job by investigating the prevalence of diabetic foot at risk of ulcer development, which has contributed to knowledge by being the first study.

There are only very few typographical errors

1 – line 73: "y" should change to "and"

2 – line 80: "y" should also change to "and"

3 – line 115: (TMV y LFPM) should change to “(TMV and LFPM)”

4 – line 138: (TMV y LFPM) should change to “(TMV and LFPM)”

5- line 139: “synthetized” should be changed to “synthesized”

6 – line 242: “on Africa” should change to “in Africa”

7 – line 252: “prevalence de 44.3%” should change to “prevalence of 44.3%”

8 – line 401:” to stablish” should change to “to establish”

The manuscript described a technically sound piece of scientific research with data that supports the conclusions. They used appropiate tools in analysing their data

6. PLOS authors have the option to publish the peer review history of their article (what does this mean?). If published, this will include your full peer review and any attached files.

Reviewer #1: **Yes: **Vincent Pam Gyang

---

## [Author Response · Author response to Decision Letter 0]

2 Mar 2023

Dear reviewers, a response is given to each of the observations

Reviewer 1

COMMENTS

The authors have done a good job by investigating the prevalence of diabetic foot at risk of ulcer development, which has contributed to knowledge by being the first study.

There are only very few typographical errors

1 – line 73: "y" should change to "and"

Corrected

2 – line 80: "y" should also change to "and"

Corrected

3 – line 115: (TMV y LFPM) should change to “(TMV and LFPM)”

Corrected

4 – line 138: (TMV y LFPM) should change to “(TMV and LFPM)”

Corrected

5- line 139: “synthetized” should be changed to “synthesized”

Corrected

6 – line 242: “on Africa” should change to “in Africa”

Corrected

7 – line 252: “prevalence de 44.3%” should change to “prevalence of 44.3%”

Corrected

8 – line 401:” to stablish” should change to “to establish”

Corrected

 

ACADEMIC EDITOR

introduction 

Does the first sentence refer to worldwide estimates? To any diabetes?

I do not understand the sentence towards the end of para 1 " Moreover, the healthcare costs for diabetic foot treatment equals costs related to oncological fields, being a burden for developing countries " – what does that mean? please clarify with some statistical data? 

Answer

We modified the introduction according to the suggestions.

Methods

 The authors say that if there is significant heterogeneity they will conduct a subgroup analysis by continents, sex, type of study, type of DM, type of population, age group, time of DM and history of CKD was performed. What is the a priori justification for selecting these variables as potential sources of heterogeneity?

Answer

We modified methods according to the suggestions.

Results

In general, I found it hard to wade through the results. Virtually all numbers presented in tables are reiterated in the text - this is unnecessary and I would recommend only the relevant results are emphasised with numbers.

The data seem to show important regional differences e.g. prevalence rates seems to be very high in the South America and Central America (SACA) presented the highest prevalence with the lowest prevalence was identified on Africa compared to other regions. These are important findings that need discussion in terms of implications in practices and research in particular in limited resources countries in those regions.

It was virtually impossible to read the forest plans

Answer

We modified the results according to the suggestions.

wWe improve the resolution of the figures

Discussion

 I request that authors expand the discussion to be more narrative and also forward-looking and also including a section on directions of future research.

Answer

We modified the discussion according to the suggestions and added a paragraph on future research.

“Given the multiple forms of evaluation of neuropathy. Comparing possible versus probable neuropathy according to the Toronto Consensus in ulcer development is important. Likewise, we generally find 30% arterial calcification in the ABI that limits the determination of flow. Therefore, we must validate adding the evaluation of the arterial waveform by portable vascular Doppler for ulcer occurrence. 

Carrying out studies based on the hospital population is more feasible. But we should know the frequency of foot at risk in the general population, where the prevalence is lower, and we find people with better diabetes control or less access to health services.”

Conclusions

 Weak conclusion which says little - can you say something more concrete

Answer

We modified the conclusion according to the suggestions.

“The overall prevalence of foot at risk is high on worldwide. We identified high between-study heterogeneity and significant limitations (e.g limited countries, heterogeneous definitions, hospital-based studies, low certainty level of evidence). We need upgraded research using standardized and population-based studies and urgent action against preventable causes of diabetic foot.”

 

Edriam Nim Tolentino

1. Please provide additional details regarding participant consent. In the Methods section, please ensure that you have specified (1) whether consent was informed and (2) what type you obtained (for instance, written or verbal). If your study included minors, state whether you obtained consent from parents or guardians. If the need for consent was waived by the ethics committee, please include this information. 

Our study was a systematic review of published studies. It was not required to obtain informed participation from people. However, we request authorization from an ethics committee authorized in Peru.

In the manuscript

“ This research did not include people, we only evaluated published studies. It was not necessary to require participan consent. We obtained authorization from the Institutional Ethics and Research Committee of thethrough the certificate 122-CIEI-CIENTIFICA-2021”

2. We note that Figure 1 in your submission contain [map/satellite] images which may be copyrighted. All PLOS content is published under the Creative Commons Attribution License (CC BY 4.0), which means that the manuscript, images, and Supporting Information files will be freely available online, and any third party is permitted to access, download, copy, distribute, and use these materials in any way, even commercially, with proper attribution. For these reasons, we cannot publish previously copyrighted maps or satellite images created using proprietary data, such as Google software (Google Maps, Street View, and Earth). For more information, see our copyright guidelines: http://journals.plos.org/plosone/s/licenses-and-copyright.

Figure 1 is an editable diagram of selection of included studies. The developers themselves require its use in systematic reviews. We place the citation and permission for its use granted by the developer.

From: Page MJ, McKenzie JE, Bossuyt PM, Boutron I, Hoffmann TC, Mulrow CD, et al. The PRISMA 2020 statement: an updated guideline for reporting systematic reviews. BMJ 2021;372:n71. doi: 10.1136/bmj.n71. The PRISMA Statement and the PRISMA Explanation and Elaboration document are distributed under the terms of the Creative Commons Attribution License, which permits unrestricted use, distribution, and reproduction in any medium, provided the original author and source are credited. (http://prisma-statement.org/PRISMAStatement/CitingAndUsingPRISMA.aspx

We replace figure 2 with another image with the requested license cc by 4.0

“Continental organizations” by Sbb1413 is licenced under CC-BY 4.0 /Modified from original.

---

## [Editor Report · Decision Letter 1]

22 Mar 2023

Prevalence of diabetic foot at risk of ulcer development and its components stratification according to the International Working Group on the Diabetic Foot (IWGDF): A systematic review with metanalysis

PONE-D-22-29064R1

Dear Dr. Marlon Yovera-Aldana

We’re pleased to inform you that your manuscript has been judged scientifically suitable for publication and will be formally accepted for publication once it meets all outstanding technical requirements.

Kind regards,

Mahmoud M Werfalli, PhD

Academic Editor

PLOS ONE